# Quantitative Trait Loci for Heat Stress Tolerance in *Brassica rapa* L. Are Distributed across the Genome and Occur in Diverse Genetic Groups, Flowering Phenologies and Morphotypes

**DOI:** 10.3390/genes13020296

**Published:** 2022-02-03

**Authors:** Sheng Chen, Alice Hayward, Shyam S. Dey, Mukesh Choudhary, Khaing P. Witt Hmon, Fabian C. Inturrisi, Aria Dolatabadian, Ting Xiang Neik, Hua Yang, Kadambot H. M. Siddique, Jacqueline Batley, Wallace A. Cowling

**Affiliations:** 1The UWA Institute of Agriculture, The University of Western Australia, Perth, WA 6001, Australia; shyam.iari@gmail.com (S.S.D.); mukesh.choudhary@research.uwa.edu.au (M.C.); khaingpann@gmail.com (K.P.W.H.); kadambot.siddique@uwa.edu.au (K.H.M.S.); jacqueline.batley@uwa.edu.au (J.B.); wallace.cowling@uwa.edu.au (W.A.C.); 2School of Agriculture and Environment, The University of Western Australia, Perth, WA 6001, Australia; 3Queensland Alliance for Agriculture and Food Innovation, The University of Queensland, St Lucia, Brisbane, QLD 4072, Australia; a.hayward@uq.edu.au; 4Division of Vegetable Science, ICAR-Indian Agricultural Research Institute, New Delhi 110012, India; 5School of Biological Sciences, The University of Western Australia, Perth, WA 6001, Australia; fabian.inturrisi@hotmail.com (F.C.I.); aria.dolatabadian@umanitoba.ca (A.D.); tingxiang@gmail.com (T.X.N.); yanghua.northwest@gmail.com (H.Y.)

**Keywords:** field mustard, sarson, turnip, Chinese cabbage, canola, heat stress tolerance, controlled environment, QTL

## Abstract

Heat stress events during flowering in *Brassica* crops reduce grain yield and are expected to increase in frequency due to global climate change. We evaluated heat stress tolerance and molecular genetic diversity in a global collection of *Brassica rapa* accessions, including leafy, rooty and oilseed morphotypes with spring, winter and semi-winter flowering phenology. Tolerance to transient daily heat stress during the early reproductive stage was assessed on 142 lines in a controlled environment. Well-watered plants of each genotype were exposed to the control (25/15 °C day/night temperatures) or heat stress (35/25 °C) treatments for 7 d from the first open flower on the main stem. Bud and leaf temperature depression, leaf conductance and chlorophyll content index were recorded during the temperature treatments. A large genetic variation for heat tolerance and sensitivity was found for above-ground biomass, whole plant seed yield and harvest index and seed yield of five pods on the main stem at maturity. Genetic diversity was assessed on 212 lines with 1602 polymorphic SNP markers with a known location in the *B. rapa* physical map. Phylogenetic analyses confirmed two major genetic populations: one from East and South Asia and one from Europe. Heat stress-tolerant lines were distributed across diverse geographic origins, morphotypes (leafy, rooty and oilseed) and flowering phenologies (spring, winter and semi-winter types). A genome-wide association analysis of heat stress-related yield traits revealed 57 SNPs distributed across all 10 *B. rapa* chromosomes, some of which were associated with potential candidate genes for heat stress tolerance.

## 1. Introduction

*Brassica napus* (oilseed rape, canola), a globally important oilseed crop, is vulnerable to heat and drought stress, especially during the early reproductive stage. *B. napus* has a relatively narrow gene pool due to its recent evolution from its ancestors *B. rapa* (field mustard, turnip, sarson, Asian cabbage) and *B. oleracea* (Mediterranean cabbage) [1], and as a result of ‘bottlenecks’ during original polyploidisation events, subsequent agricultural selection in spring and winter pools, selection for canola quality and breeding in isolated regional environments [2]. This situation was accentuated in Australia, where selection in a closed spring canola breeding population significantly reduced genetic diversity over 30 years (1970–2000) due to genetic drift [3]. Consequently, research has focused on the ancestors of *B. napus* (such as *B. rapa*) as potential sources of heat tolerance [4].

*B. rapa* is a traditional oilseed and vegetable crop that has been cultivated globally for more than 6000 years [5]; it has a wide geo-distribution with some types flourishing in heat and drought-affected regions [6]. The species includes a diverse range of agricultural morphotypes (leafy, rooty and oilseed types) and phenologies (winter, spring and semi-winter types) which arose after initial domestication from wild types in Central Asia [7] followed by independent selection in Europe and Asia [7,8,9,10,11,12].

*B. napus*, *B. rapa* and *B. juncea* were very sensitive to heat stress during flowering, which decreased subsequent pod and seed formation [13,14]. Temperatures greater than 29.5 °C during flowering in the field decreased seed yield in *B. napus* [15]. Young et al. [16] showed that high-temperature stress in *B. napus* (35 °C for 4 h each day for 1 or 2 weeks after the initiation of flowering) reduced fruit and seed development, pollen germination and in vivo pollen tube growth. Using similar transient temperature stress for 7 days after the first open flower, Annisa et al. [4] found genetic variation in pod and seed number following heat stress in six spring-type *B. rapa* accessions. Chen et al. [17] found that transient heat stress negatively affected male and female reproductive organs of *B. napus,* with female organs more sensitive than male organs to heat stress.

Based on our previous experience that heat stress has its greatest impact during the reproductive stage in *B. rapa* and *B. napus* [4,13,14,17], we evaluated a genetically diverse global collection of *B. rapa* germplasm for heat stress tolerance following simulated transient daily heat stress after the first open flower on the main stem of well-watered plants. We included a range of morphotypes (leafy, rooty and oilseed types) and flowering phenologies (spring, winter and semi-winter types). We assessed the genotypic diversity of the population using SNP markers. We used a genome-wide association analysis (GWAS) to investigate significant quantitative trait loci (QTL) for heat tolerance and potential relationships between heat tolerance and geographic origin, morphotype and/or flowering phenology. We searched for candidate genes associated with significant SNPs which may contribute to heat tolerance.

## 2. Materials and Methods

### 2.1. Plant Materials

A total of 217 *B. rapa* lines were used in this study (Appendix A). These accessions were chosen based on their self-compatibility reported in Annisa et al. [10] and Guo et al. [12] and represent a wide range of genetic diversity groups, morphotypes and flowering phenologies. Of these, 75 lines with a limited number of seeds and with self-incompatibility were excluded for phenotyping experiment, so 142 lines were phenotyped for several physiological and yield-related traits.

### 2.2. Temperature Treatments

Two sets of plants were sown in pots, with a single plant in each pot. Set 1 plants were grown for the control (no heat stress) treatment and Set 2 plants were grown for the heat stress treatment at first flower on the main stem. There were three replications (pots) of each line in each set. Plants were grown in 8.1 L pots 230 mm in depth (Garden City Plastics, Perth, Australia) in a greenhouse at The University of Western Australia, Crawley, Western Australia (31°57′ S, 115°47′ E) with an average relative humidity of 65%, 25°C/15°C (day/night) temperature and average light intensity 635 mmol m^−2^ s^−1^ PAR under optimal moisture conditions. Light readings in the glasshouse were taken at mid-day on a sunny day in mid-spring in an unshaded area of the glasshouse. Each pot contained 4.5 kg of canola potting mix, comprising 50% fine composted pine bark, 20% coco peat and 30% brown river-sand plus 1.0 g of gypsum per kg with its final pH of ~6.0. After sowing, each pot was fertilised every two weeks with 0.25 g of soluble nutrient powder (Thrive, Yates Australia, NSW, Australia; containing 25% N, 5.0% P, 8.8% K, 4.6% S, 0.5% Mg, 0.18% Fe, 0.01% Mn, 0.005% Cu, 0.004% Zn and 0.001% Mo). When the first open flower was seen on the main stem, plants were moved to a controlled environment room (CER) for seven days of temperature treatment, and the first five floral buds on the main stem (flowers opened on the first day) were tagged. Set 1 plants were transferred to CER1 (control treatment) and Set 2 plants to CER2 (heat stress treatment). After 7 d of temperature treatment, the plants were returned to the greenhouse and maintained with adequate nutrition and moisture until seed harvest at maturity.

Both CER1 and CER2 were set to 16 h day and 8 h night with 425 mmol m^−2^ s^−1^ light and 65% relative humidity. CER1 (control) was set at 25 °C day/15 °C night. CER2 (heat stress) was set at 25 °C night, and during the day the temperature was gradually increased in the first 6 h to 35 °C, maintained at 35 °C for 4 h and then gradually decreased over the next 6 h to 25 °C. Plants in both treatments were watered regularly through low-flow (35 mL h^−1^) drippers to maintain the soil water content at not less than 90% field capacity. Both high temperature and control temperature CERs had the same light intensity, same day-length, same humidity, same nutrition and adequate water to avoid any confounding of heat stress treatment with other effects that might have differed between the treatments.

Both sets of plants, destined for high and control temperature treatments, received the same growth conditions before, during and after the temperature treatments to avoid any confounding of heat stress with other growth inputs such as water, nutrients and light.

### 2.3. Phenotyping

Several physiological traits were measured between 11.00 and 13.00 h on the seventh day of temperature treatment (Table 1). The temperature of a recently opened flower and a nearby leaf was measured using an infrared thermometer (Impac^®^ Model IN 15 plus, LumaSense Technologies GmbH, Santa Clare, CA, USA) with a minimum 2.2 mm diameter measurement area. A separate digital thermometer with a 1 s response time measured the ambient temperature. The leaf and bud temperatures and the ambient temperature were recorded simultaneously with four repeat measurements per leaf and bud. The temperature difference between bud and ambient environment (T1, °C), leaf and ambient environment (T2, °C) and bud and leaf (T3, °C) were calculated. Stomatal conductance of the youngest fully expanded leaf was measured using an SC1 leaf porometer (Decagon Devices, Pullman, WA, USA) on adaxial and abaxial leaf surfaces. Leaf conductance (LC, mmol m^−2^ s^−1^) was the sum of the adaxial and abaxial conductance. The chlorophyll content index (CI, SPAD unit) was measured on intact leaves with a portable chlorophyll meter SPAD-502Plus (Konica Minolta Sensing Americas, Ramsey, NJ, USA). CI is the ratio of the leaf transmittance in red light at 650 nm (at which chlorophyll absorbs) and in near-infrared light at 940 nm (for the correction of leaf thickness).

Several yield-related traits were also measured at maturity. Each plant was cut at the soil level and dried at 32 °C for 14 d before measuring the above-ground biomass (BM, g). The seed pods from each plant were harvested at maturity, threshed manually and cleaned using a vacuum separator (Kimseed, Wangara, WA, Australia). Seed yield from the first five floral buds on the main stem (Y5P, g) and seed yield on each whole plant (YWP, g) were measured. The harvest index (HI, %) was calculated based on the ratio of YWP and BM for each plant.

Line performance in the control treatment was identified by the suffix ‘_C’ (e.g., BM_C), and line performance in the heat stress treatment was identified by the suffix ‘_H’ (e.g., BM_H).

Line performance was also compared across C and H treatments. Line performance in control $\left(Y_{c}\right)$ and heat stress $\left(Y_{h}\right)$ treatments, and the mean performance of all lines in the control treatment $({\bar{Y}}_{c}),$ were used to calculate the stress tolerance index (STI) [18] (Equation (1)) and percentage change from the control to heat stress treatment (%C) (Equation (2)) as follows:(1)$$STI=\frac{\left(Y_{c}\times Y_{h}\right)}{{\bar{Y}}_{c}{}^{2}}$$
(2)$$\%C=\frac{100\left(Y_{h}-Y_{c}\right)}{Y_{c}}$$

### 2.4. Statistical Analysis of Phenotypic Traits

Statistical analysis of phenotypic traits was based on linear mixed models with residual maximum likelihood estimation in ASReml-R version 4 [19]. For all the measured traits, the temperature treatment (‘Treatment’) was considered a fixed effect and Line, Line × Treatment and Error as random effects. The significance of fixed effects was evaluated by Wald statistic and random effects by Z statistic of variance components. Principal component analysis (PCA) biplot and clustering were performed using the ‘factoextra’ and ‘FactoMineR’ packages in R Version 4.0.2.

### 2.5. SNP Genotyping, Population Structure and Genetic Diversity Analysis

Leaf samples were combined from three plants at the four-leaf stage for each *B. rapa* line to extract genomic DNA using the Qiagen DNeasy plant kit (Düsseldorf, Germany). SNP genotyping was assessed on a *B. napus* Illumina Infinium 6K array described in Dalton-Morgan et al. [20]. Chips were scanned using an Illumina HiScan SQ machine. Samples with their call rate below 70% were excluded from further data analysis. SNP locations were established by BLASTing to the *B. rapa* genome sequence (version 1.5) [21]. All SNPs located on the *B. rapa* A genome with amplification in >90% of experimental lines and a minor allele frequency >10% across the diversity set were selected for further analysis.

The software package STRUCTURE version 2.3.4 [22] was used to detect population structure [23]. Kinship (K) was calculated with SPAGeDi 1.3 [24,25]. A hierarchical cluster analysis was performed using the unweighted pair group method with arithmetic averages (UPGMA) as proposed by Sneath and Sokal (1973), and the ordination analysis was performed using PRIMER 6 software [26].

### 2.6. Genome-Wide Association Mapping

A general linear model (GLM) and a mixed linear model (MLM) were compared in TASSEL [27] to identify QTL significantly associated with yield and heat tolerance traits and to assess the effect of the kinship matrix for family relatedness estimates (K), principal component analysis (P) and population structure (Q). We compared three different models in GLM: (1) the naive model (no control for kinship or population structure), (2) the Q model (controls for population structure) and (3) the P model (controls for principal components) with the top 10 principal components included as fixed effects [28,29]. We also compared three different models in MLM: (1) a model based on kinship (K), (2) a model that unified population structure and kinship (Q + K) and (3) a model based on principal component and kinship (P + K). The *p*-values obtained from all models were converted into −log10 (*p*). The variation of observed −log10 (*p*) for each SNP from marker-trait associations was compared with the expected *p*-values in quantile-quantile (QQ) plots. We evaluated these six models for false positives and false negatives based on the QQ plots. A sharp deviation from the expected *p*-value distribution in the tail area would indicate that a model appropriately controlled both false positives and false negatives [30]. The significance of the association between markers and traits was based on the threshold −log10 (*p*) = 3.0.

### 2.7. Candidate Gene Discovery

The flanking SNPs of each QTL were used to blast the reference genome sequence of *B. rapa* (version 1.5) [21] and the A genome of *B. napus* using the Darmor-bzh genome reference [1] to locate the position of SNPs. Genome annotation files were downloaded from NCBI (https://www.ncbi.nlm.nih.gov/, accessed on 8 January 2021). All genes within 100 bp of the QTL were selected for further analysis. The gene function and other information were evaluated, including the Kyoto Encyclopedia of Genes and Genomes (KEGG) Orthology, gene ontology (GO) component, GO function and GO process.

## 3. Results

### 3.1. Phenotypic Variation and Heat Stress Tolerance

Of the 142 *B. rapa* lines observed for five physiological traits measured on day 7 of the temperature treatment and four yield-related traits measured at maturity, eight lines showed self-incompatibility and were excluded from the analysis.

The main effect of temperature treatment (‘Treatment’) was highly significant for all nine traits measured (Table 2), indicating that heat stress significantly increased or decreased the mean value of the trait across all lines relative to the control treatment. For example, the percent change (%C) under heat stress relative to the control in YWP, HI and Y5P, averaged across all lines, was –39.09%, –50.03% and –98.08%, respectively, while %C in BM was +8.64% under heat stress (Table 2). The %C in T3, CI and LC were +85.50%, +5.28% and +73.47%, respectively; that is, the high-temperature treatment enhanced the physiological activities of the plants in the absence of moisture and nutritional stress and increased the accumulation of dry matter (BM). However, heat stress inhibited the reproductive process and %C for seed yield (both YWP and Y5P), and HI was negative (Table 2).

There were significant or highly significant interaction effects of Line × Treatment for all nine traits measured (Table 2), suggesting that the performance rank of these 134 *B. rapa* lines varied in the control and heat stress treatment for all nine traits.

PCA was based on %C and STI for Y5P, YWP, BM and HI in 134 lines (Figure 1). The first principal component (PC1) explained 30.61% of the variance and was strongly associated with %C and STI of YWP and HI. The second component (PC2) explained 19.67% of the variance and was negatively associated with Y5P_STI and Y5P_%C. PC2 separated six lines, particularly R101 and R090, with relatively higher YWP and Y5P under heat stress.

Cluster analysis, also based on %C and STI for Y5P, YWP, BM and HI in 134 lines, revealed two major clusters, heat-tolerant (Tol) and heat-sensitive (S), each of which had two sub-clusters (Tol1, Tol2 and S1, S2) (Figure 2). Tol1 and Tol2 had an average %C in YWP of –8.38% and –22.81%, respectively, while S1 and S2 had average %C in YWP reduction of –53.67% and –42.48%, respectively (Table 3). Tolerant groups showed a greater increase in BM under heat stress than sensitive groups and experienced less reduction in HI. Tolerant groups had higher STI (higher overall performance) for HI than sensitive groups. Tol2 suffered the least impact of heat stress on Y5P (yield of five pods on the main stem) (Table 3). One line, R101 (a yellow sarson from India), had relatively high YWP and Y5P under heat stress (Figure 1, Appendix A).

Five physiological traits measured on the 7th day of temperature treatment also showed a similar trend (Table 3). The tolerant groups had cooler buds (T1) and leaves (T2) than the sensitive groups, and Tol2 had the coolest buds and leaves. The temperature difference between bud and leaf (T3) in heat-tolerant lines also showed higher %C than in the heat-sensitive lines (Table 3), consistent with our previous reports [13,14]. Tolerant groups had better photosynthesis and transpiration performance than heat-sensitive groups, as reflected by the relatively higher chlorophyll content index and leaf conductance (Table 3).

### 3.2. SNP Genetic Diversity and Genetic Population Structure versus Morphotype, Flowering Phenology and Heat Tolerance and Sensitivity

All 217 *B. rapa* lines were subjected to SNP genotyping, and the average call rate of all lines was 87.04%, and five lines with their call rate below 70.0% were excluded from further data analysis. A total of 1602 SNP alleles were polymorphic across 212 lines, and these SNPs were used in a population structure analysis. The optimum solution in STRUCTURE was four genetic sub-populations (Figure 3 and Figure 4), and this was supported by two major populations each with two sub-populations in a phylogenetic cluster analysis based on PRIMER 6 (Figure 5).

These four genetic sub-populations reflected the geographical origin of lines: A1 from East Asia (China, South Korea, North Korea and Japan), A2 from South Asia (India, Pakistan, Afghanistan, Bangladesh and Nepal) and B1 and B2 from mainly European countries. Various morphotypes (leafy, rooty and oilseed types) were distributed across each of the four sub-populations, as were winter, semi-winter and spring types (Figure 5 and Appendix A). The heat-tolerant lines in Tol1 and Tol2 and sensitive lines in S1 and S2 were also randomly distributed across the four genetic sub-populations (Figure 5).

### 3.3. Genome-Wide Association Analysis and Candidate Genes Related to Heat Stress Tolerance

In order to remove or reduce skewness and normalise the distribution of the data, some phenotypic traits were transformed before the marker-trait association analysis. Log transformation was applied to Y5P_C, HI_C, Y5P_H, YWP_H and HI_H, and cube root transformation was applied to BM_C and YWP_C.

After comparing six models, an MLM based on population structure and kinship Q+K was selected to identify QTL significantly associated with yield and heat tolerance traits. The MLM(Q+K) model showed a relative sharper deviation from the expected P-value distribution in the tail area for all yield-related traits (Appendix A).

A total of 57 SNP markers were significantly associated with at least one of the yield-related traits with a logarithm of odds (LOD) score greater than 3.0 (Table 4). Five SNPs were associated with YWP_%C and distributed on four chromosomes (A03, A04, A05 and A07). A single SNP, UQnapus5640, was associated with YWP_STI at LOD 4.37 and explained 19.06% of the phenotypic variance (Table 4).

Six SNPs were associated with traits under control and heat stress conditions and distributed on six chromosomes: UQnapus1557 for BM_C and BM_H (Chromosome A03), UQnapus1923 for HI_C and Y5P_H (Chromosome A04), UQnapus2563 for HI_C and HI_H (Chromosome A05), UQnapus2473 for Y5P_H and YWP_C (Chromosome A07), UQnapus2777 for HI_C and YWP_C (Chromosome A09) and UQnapus4615 for BM_C and BM_H (Chromosome A10) (Table 4).

Among the 57 significant SNPs, the range of phenotypic variance explained ranged from 6.52% for UQnapus1680 in relation to BM_H to 23.43% for UQnapus3633 in relation to HI_C. Most SNPs explained <10% of the total phenotypic variance, indicating that heat tolerance is a complex quantitative trait, with several QTLs for each trait.

Under heat stress conditions, 24 SNPs were associated with seed yield-related traits and distributed on all 10 *B. rapa* chromosomes (Table 4). Under control conditions, 33 SNPs were associated with seed yield-related traits and distributed on all 10 chromosomes (Table 4).

One SNP marker, UQnapus2563 located on Chromosome A05, was associated with two traits. This marker explained 20.35% of the phenotypic variance for HI_C with an LOD of 6.31 and 15.9% of the phenotypic variance of HI_H with an LOD of 4.58 (Table 4). A search of the A genome of *B. rapa* and *B. napus* in a 230 kb genomic region flanking UQnapus2563 revealed 49 candidate genes present in both species (Appendix A).

Another SNP marker, UQnapus2473 located on Chromosome A07, was associated with Y5P_H at an LOD of 4.69 and explained 15.67% of the phenotypic variance. UQnapus2473 was also associated with YWP_C at an LOD of 3.71, explaining 10.81% of the phenotypic variance (Table 4). A search of the genomic region of 225 kb flanking UQnapus2473 found nine candidate genes in the *B. napus* A genome, eight of which were also in the *B. rapa* genome (Table 5). These nine candidate genes were further searched for orthologs in *Arabidopsis thaliana.* Function annotation revealed that several candidate genes, including the protein kinase gene AT2G18890.1 and several C2H2 zinc finger protein genes, were related to the abiotic stress response. Candidate gene AT2G18510.1 is involved in embryo development ending in seed dormancy. Candidate gene AT2G18500.1 encodes ovate family protein 7 (OFP7) located in the plasma membrane and expressed in embryos, flowers and seeds (Table 5).

## 4. Discussion

Heat tolerance is an important trait for *Brassica* grain crops exposed to high temperatures during flowering to avoid grain yield losses due to heat stress [13,14,15,16]. Heat stress events during crop flowering are expected to increase in frequency in the future due to global warming [31]. This study simulated heat stress during flowering in a globally diverse collection of *B. rapa* lines and compared their responses to transient daily heat stress relative to a control temperature treatment for 7 d from the first open flower. The plants were provided with adequate moisture and nutrients to avoid other stresses during the temperature treatment. As a result, several promising heat-tolerant *B. rapa* lines were identified, which maintained grain yield, biomass and harvest index after 7 d of heat stress at the first flower, while heat-sensitive lines suffered a significant reduction in yield-related traits in the heat stress treatment relative to the control.

*B. rapa* is a traditional oilseed and vegetable crop with broad genetic diversity and wide geographic distribution, with some types flourishing in heat-affected regions. In previous research, a leafy vegetable morphotype of *B. rapa* from Indonesia tolerated heat stress during flowering and early seed fill [4]. Hence, we expanded our survey of heat stress tolerance in *B. rapa* to include leafy, rooty and oilseed morphotypes with spring, winter and semi-winter flowering phenologies from diverse geographic origins. Forty-one out of 134 lines were heat tolerant and distributed across four SNP sub-populations identified from a cluster analysis based on 1602 polymorphic SNP markers with geographic origins in East Asia (A1), South Asia (A2) and Europe (B1 and B2). Heat-tolerant lines were found among all morphotypes (leafy, rooty and oilseed) and flowering phenologies (spring, winter and semi-winter types) (Figure 5).

Phylogenetic analyses of the 212 *B. rapa* lines in this study follow previous results on this population using microsatellite markers [4,12]. The various morphotypes (leafy, rooty and oilseed type) were distributed across each of the four SNP sub-populations, as were winter, semi-winter and spring flowering phenologies (Figure 5). The four sub-populations based on SNP genetic diversity reported in this study reflect the geographical origins of lines but not their morphotype or flowering phenology. This is consistent with reports [11,12] in which similar morphotypes from different regions were often not genetically related, but contrasts with Bird et al. [32] who reported five SNP sub-populations in *B. rapa,* which tended to be divided by morphotype and geographic origin.

Genome-wide association studies of heat stress-related yield traits revealed 57 SNPs, with some associated with potential candidate genes for heat stress tolerance. Identifying SNP markers and candidate genes associated with heat stress-affected traits helps select stress-tolerant accessions in breeding programs. The 57 SNPs were distributed across all 10 *B. rapa* chromosomes, with the markers explaining between 6.52% (UQnapus1680 for BM_H) and 23.43% (UQnapus3633 for HI_C) of the phenotypic variance. On average, each SNP explained a small percentage of phenotypic variance. Heat stress tolerance is a complex quantitative trait, with no major genes found in this study.

Complex quantitative genetic control of heat stress and other abiotic stress tolerance has been reported in Brassica [33] and other crop species [34]. A study of heat stress tolerance in spring-type *B. napus* under controlled conditions reported that heat stress increased pollen sterility and sterile/aborted pods and reduced pod numbers on the main stem [35] but did not investigate seed yield, biomass or harvest index.

Nine candidate genes flanking UQnapus2473 were found in the *B. rapa* genome and the *B. napus* A genome (Table 5). Function annotation of orthologs in *A. thaliana* revealed that several candidate genes, such as protein kinase gene AT2G18890.1 and several C2H2 zinc finger protein genes, were related to abiotic stress response. AT2G18510.1 is involved in embryo development, and the embryo development stage is very sensitive to heat stress in *B. napus* [14]. AT2G18500.1 encodes ovate family protein 7 located in the plasma membrane and expressed in embryos, flowers and seeds. Ovate family proteins are a family of plant growth regulators that play diverse roles in many aspects of physiological processes. Ovate family protein 1 regulates the drought stress response in *Populus trichocarpa* [36].

However, further functional confirmation research is needed to validate these candidate genes in response to heat stress in *B. rapa*. Once confirmed, breeders can potentially transfer valuable heat stress tolerance QTL from *B. rapa* to *B. napus*.

Future research should use a higher density linkage map to minimise the QTL region on the genome limiting the number of candidate genes and undertake functional characterisation of candidate genes via qRT-PCR. Eventually, it should be possible to use functional markers to assist the screening of heat-tolerant germplasm and marker-assisted introgression of multiple QTL from *B. rapa* to *B. napus* for heat tolerance.

## 5. Conclusions

*B. rapa* is a traditional oilseed and vegetable crop with broad genetic diversity and wide geographic distribution. Large genetic variation for heat tolerance and sensitivity was found for above-ground biomass, whole plant seed yield and harvest index and seed yield of five pods on the main stem at maturity. Phylogenetic analyses confirmed two major genetic populations from East/South Asia and Europe. Heat stress-tolerant lines were distributed across each population and included diverse geographic origins, morphotypes (leafy, rooty and oilseed) and flowering phenologies (spring, winter and semi-winter types). A genome-wide association analysis of heat stress-related yield traits revealed 57 SNPs distributed across all 10 *B. rapa* chromosomes, some of which were associated with potential candidate genes for heat stress tolerance. In summary, QTL for heat stress tolerance in *B. rapa* are distributed across the genome and occur in diverse genetic groups, flowering phenologies and morphotypes.

## Figures and Tables

**Figure 1 genes-13-00296-f001:**
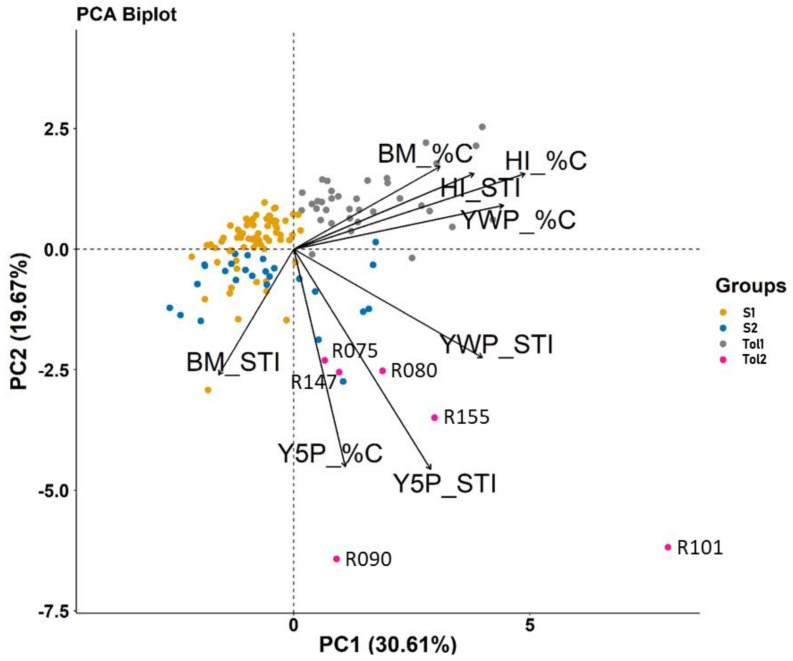
Biplot of principal component analysis (PCA) based on the percentage of change (%C) and stress tolerance index (STI) for four yield-related agronomic traits measured at maturity on 134 *B. rapa* lines under control (C) and heat stress (H) conditions: seed yield of five pods on the main stem (Y5P), above-ground biomass (BM), seed yield on whole plant (YWP) and harvest index (HI). The first two principal components PC1 and PC2 explained 30.61% and 19.67% of the variance, respectively, and separated the 134 lines into two tolerant groups (Tol1 and Tol 2) and two sensitive groups (S1 and S2). The names of six lines in Tol2 are shown.

**Figure 2 genes-13-00296-f002:**
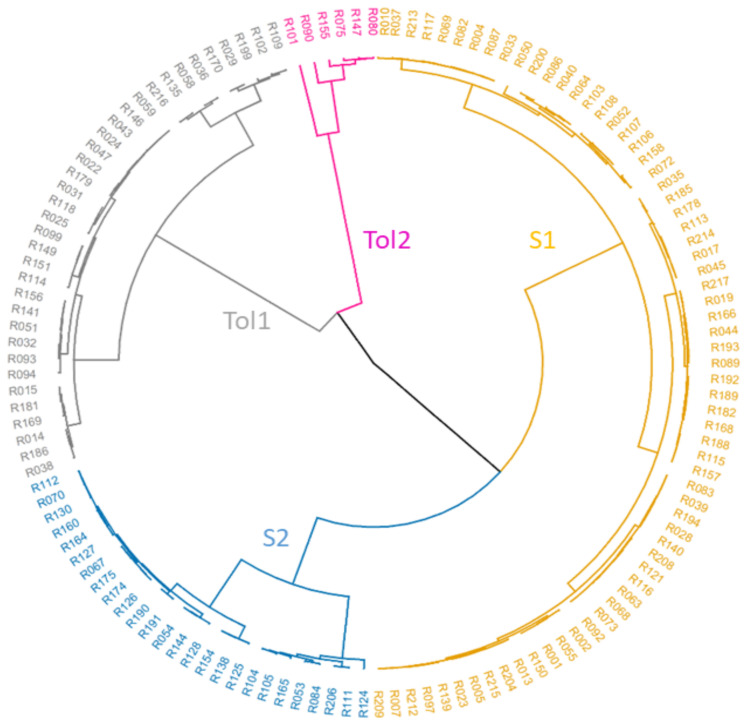
*B. rapa* lines were separated into two heat-tolerant groups (Tol1 and Tol 2) and two heat-sensitive groups (S1 and S2) based on cluster analysis of percentage change (%C) and stress tolerance index (STI) under control and heat stress conditions in four yield-related agronomic traits measured at maturity: seed yield of five pods on the main stem (Y5P), above-ground biomass (BM), seed yield on whole plant (YWP) and harvest index (HI).

**Figure 3 genes-13-00296-f003:**
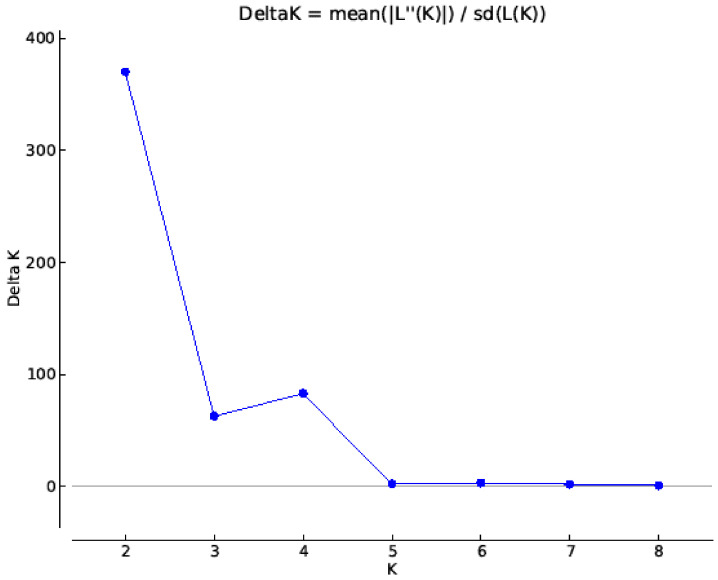
Population structure analysis showed Delta K based on the rate of LnP (D) change among successive clusters (K) from 1–8, revealing the optimum K-value for four sub-populations in the 212 *B. rapa* lines.

**Figure 4 genes-13-00296-f004:**
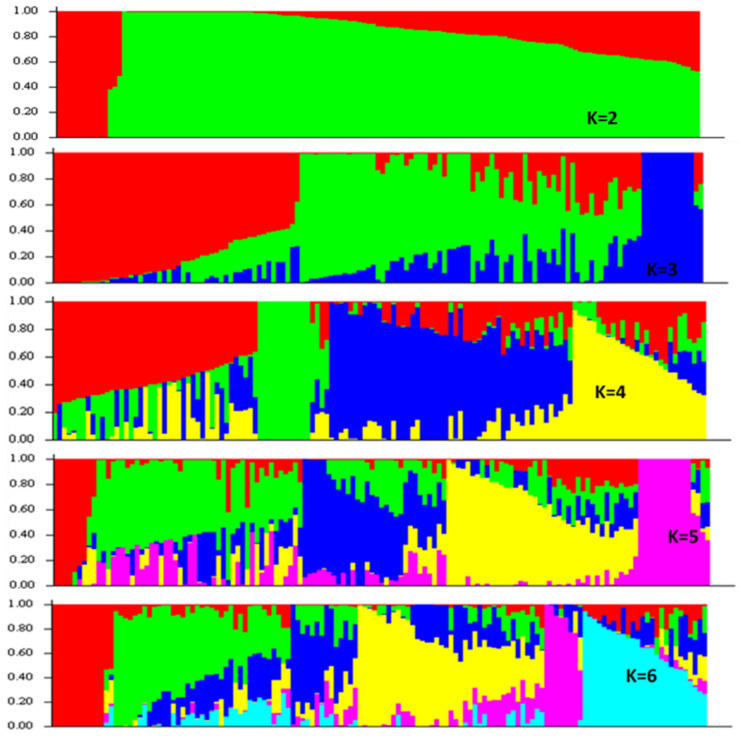
Population structure based on Q matrix structural analysis using STRUCTURE ver. 2.3.4. Four sub-populations (K = 4) were selected as the optimum outcome based on Delta K values (Figure 3) across the 212 *B. rapa* lines based on 1602 polymorphic SNPs.

**Figure 5 genes-13-00296-f005:**
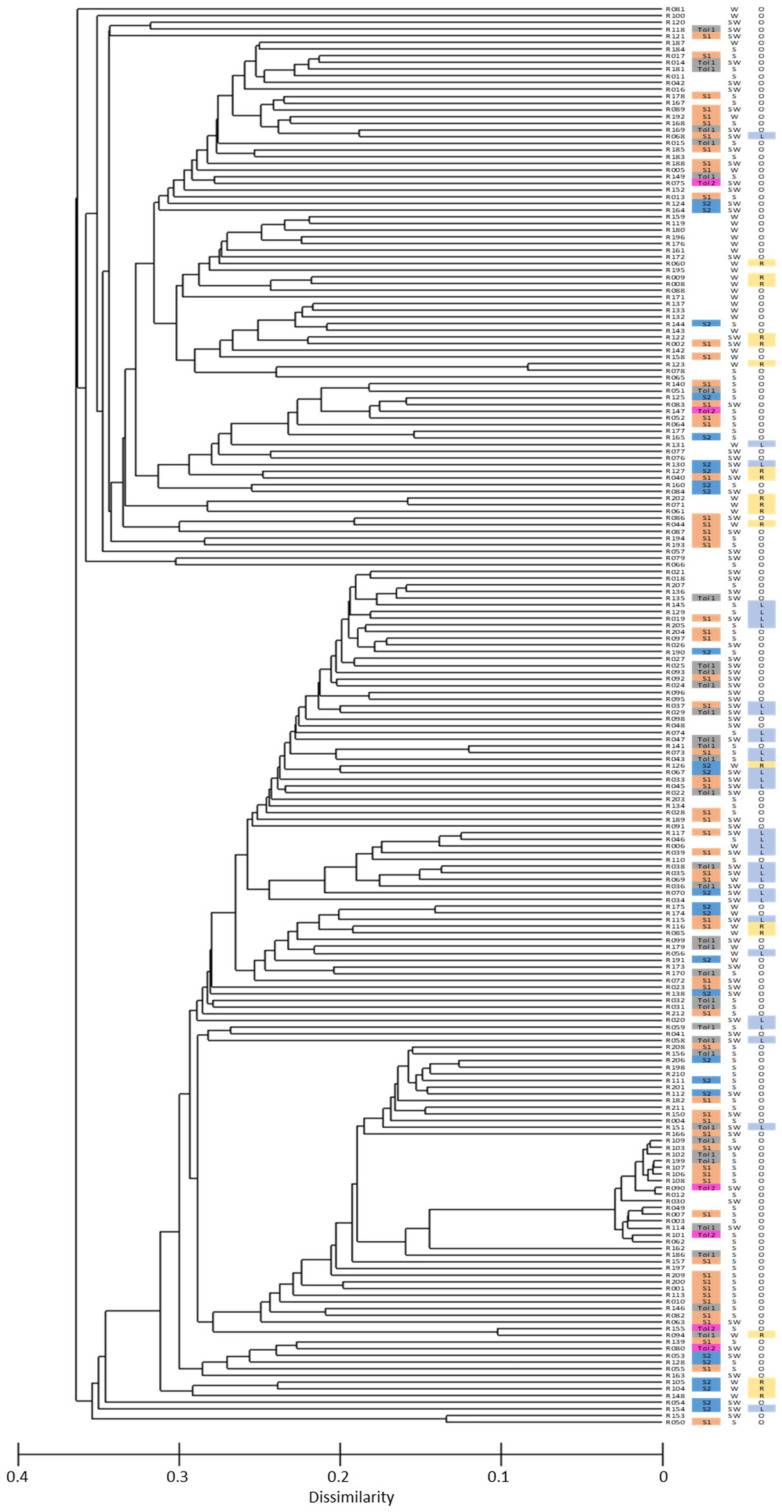
Cluster analysis of Nei’s matrix distance among 212 *B. rapa* lines based on 1602 polymorphic SNPs. Of these, 134 lines were classified into two heat-tolerant groups (Tol1 and Tol 2) and two heat-sensitive groups (S1 and S2) based on cluster analysis and principal component analysis of four yield-related agronomic traits measured at maturity (see Figure 1 and Figure 2). The flowering phenologies based on cold treatment requirements and days to flowering in the screen-house are spring (S), winter (W) and semi-winter (SW) types. The morphotypes are leafy (L), rooty (R) and oilseed (O) types.

**Table 1 genes-13-00296-t001:** List of phenotypic traits, abbreviations, units, evaluation method and stage of observation.

#	Trait	Abbreviation	Unit	Evaluation Method	Stage of Observation
1	Bud temperature depression (difference between bud and ambient temperature)	T1	°C	Leaf and bud temperatures were measured with an infrared thermometer (Impac^®^ Model IN 15 plus); a separate digital thermometer with a 1 s response time was used to measure ambient temperature.	Early flowering stage
2	Leaf temperature depression (difference between leaf and ambient temperature)	T2	°C		Early flowering stage
3	Temperature difference between bud and leaf	T3	°C		Early flowering stage
4	Leaf conductance	LC	mmol m^−2^ s^−1^	Sum of the adaxial and abaxial stomatal conductance of the youngest fully expanded leaf measured by SC1 leaf porometer.	Early flowering stage
5	Chlorophyll content index	CI	SPAD unit	Ratio of leaf transmittance in red light at 650 nm and near-infrared light at 940 nm, as measured on intact leaves with a portable chlorophyll meter SPAD-502Plus	Early flowering stage
6	Above-ground biomass	BM	g	Weight of all plant materials above the ground at harvest	Maturity
7	Seed yield whole plant	YWP	g	Weight of seed harvested from each plant	Maturity
8	Harvest index	HI	%	Ratio of seed yield vs biomass	Maturity
9	Seed yield of five pods	Y5P	g	Seed yield from five bagged floral buds on the main stem	Maturity

**Table 2 genes-13-00296-t002:** Variance components and predicted values of four seed yield-related traits measured at maturity and five physiological traits measured during heat stress at the early flowering stage of 134 *B. rapa* lines. The Line and Line × Treatment interaction effects were considered random, and the Z test was applied to test for significance. Treatment was considered a fixed effect and the Wald test was used to test for significance. The predicted value ± standard error of each trait under control (C) and heat stress (H) conditions and the percentage change (%C) under H compared to C are also provided.

Trait	Random Effects					Fixed Effects		Predicted Value		
	Line		Line × Treatment		Error	Treatment		C	H	%C
Yield-related traits at maturity										
Y5P	0.0002 ± 0.0007	^NS^	0.0065 ± 0.0009	***	0.0031 ± 0.0002	112.14	***	0.1148 ± 0.0076	0.0022 ± 0.0076	−98.08
YWP	0.1836 ± 0.0479	***	0.1913 ± 0.0421	***	0.4106 ± 0.0260	30.307	***	1.0026 ± 0.0625	0.6107 ± 0.0625	−39.09
BM	33.14 ± 4.621	***	2.289 ± 1.099	*	17.34 ± 1.095	6.85	**	10.77 ± 0.5579	11.70 ± 0.5570	8.64
HI	59.31 ± 17.43	***	33.70 ± 17.77	*	268.27 ± 17.65	24.78	***	14.11 ± 1.215	7.051 ± 1.191	−50.03
Physiological traits during heat stress										
T1	0.0001 ± 0.0000	^NS^	0.2433 ± 0.0369	***	0.1605 ± 0.0100	116.39	***	−1.496 ± 0.0475	−2.184 ± 0.0472	−45.99
T2	0.0787 ± 0.0691	^NS^	0.5825 ± 0.08879	***	0.3770 ± 0.0235	234.70	***	−2.501 ± 0.0770	−4.082 ± 0.0768	−63.21
T3	0.1133 ± 0.0455	**	0.2303 ± 0.0500	***	0.4778 ± 0.0300	124.61	***	1.007 ± 0.0621	1.868 ± 0.0616	85.50
CI	42.09 ± 6.684	***	14.96 ± 2.791	***	21.65 ± 1.351	9.22	**	33.36 ± 0.6948	35.12 ± 0.6936	5.28
LC	5365.07 ± 5546.58	^NS^	52,488.21 ± 7164.96	***	16,796.12 ± 1048.64	165.86	***	515.91 ± 21.82	894.97 ± 21.79	73.47

^1^ Phenomic traits include five physiological traits measured at early flowering during the 7 d heat stress treatment: Bud (T1) and leaf (T2) temperature depression, temperature difference between bud and leaf (T3), leaf conductance (LC), and chlorophyll content index (CI); and four yield-related agronomic traits measured at maturity: seed yield of five pods on the main stem (Y5P), above-ground biomass (BM), seed yield on whole plant (YWP), and harvest index (HI). ^2^ Statistical significance is shown as 0.01 < *p* < 0.05 (*), 0.001 < *p* < 0.01 (**) and *p* < 0.001 (***). ‘NS’ indicates ‘not significant’.

**Table 3 genes-13-00296-t003:** Summary of mean percentage change (%C) and stress tolerance index (STI) of four clustering groups (S1, S2, Tol1, Tol2) identified based on the performance of all nine phenotypic traits, including four yield-related agronomic traits measured at maturity: seed yield of five pods on the main stem (Y5P), above-ground biomass (BM), seed yield on whole plant (YWP) and harvest index (HI) on the whole plant; and five physiological traits measured at early flowering during the 7 d heat stress treatment: bud (T1) and leaf (T2) temperature depression, temperature difference between bud and leaf (T3), leaf conductance (LC) and chlorophyll content index (CI).

Trait	S1 (67 Lines)		S2 (26 Lines)		Tol1 (35 Lines)		Tol2 (6 Lines)	
	%C	STI	%C	STI	%C	STI	%C	STI
Yield-related traits at maturity								
Y5P	−98.871	0.016	−99.370	0.014	−99.536	0.015	−82.303	0.481
YWP	−53.673	0.342	−42.482	1.410	−8.375	0.968	−22.807	1.703
BM	7.447	0.908	4.687	3.591	22.798	0.638	15.532	0.799
HI	−60.615	0.274	−58.859	0.334	−37.822	1.756	−43.814	1.045
Physiological traits during heat stress								
T1	−50.850	1.472	−49.365	1.443	−60.430	1.523	−65.560	1.540
T2	−63.507	1.682	−63.447	1.603	−86.770	1.604	−98.057	1.637
T3	111.502	2.088	128.260	1.862	140.279	1.787	152.271	1.830
CI	6.581	1.036	1.135	1.334	10.004	0.995	16.235	1.174
LC	82.134	1.794	56.748	1.699	128.071	1.703	86.578	1.934

**Table 4 genes-13-00296-t004:** Fifty-seven SNP markers were associated with at least one yield-related trait with the LOD ≥ 3.0 under control (C) and/or heat stress (H) conditions. The four yield-related traits are above-ground biomass (BM), seed yield whole plant (YWP), harvest index (HI) and seed yield of five pods (Y5P). The stress tolerance index (STI) and percentage change (%C) based on YWP are considered for heat tolerance. The LOD value calculated by –log_10_ (*p*) of each associated SNP marker and the phenotypic variance explained are also listed. Markers which were associated with two or more traits are in bold and italic format.

Chromosome	Position	Marker	Trait Associated	LOD	Phenotypic Variance Explained (%)
STI and %C based on YWP					
A03	5622193	UQnapus1553	YWP_%C	3.81	15.11
A04	14473945	UQnapus5039	YWP_%C	3.26	9.45
A05	7032029	UQnapus2055	YWP_%C	3.57	12.10
A05	21539755	UQnapus5640	YWP_STI	4.37	19.06
A07	9559418	UQnapus0132	YWP_%C	3.55	10.98
A07	13661428	UQnapus0012	YWP_%C	4.56	13.26
Under heat stress condition					
A01	23298355	UQnapus5153	HI_H	3.72	8.38
A01	24614901	UQnapus1279	BM_H	3.71	14.18
A02	11002571	UQnapus1420	BM_H	3.42	10.61
A02	25794385	UQnapus1090	HI_H	5.06	15.72
A03	5898488	** *UQnapus1557* **	BM_H	4.30	12.85
A03	9882877	UQnapus2837	HI_H	3.26	12.29
A03	18737164	UQnapus1661	Y5P_H	6.07	19.61
A03	19296951	UQnapus1669	BM_H	3.24	10.96
A03	20865276	UQnapus1680	BM_H	3.12	6.52
A03	29591540	UQnapus5276	BM_H	3.05	9.98
A04	11451264	** *UQnapus1923* **	Y5P_H	3.07	11.44
A04	18294405	UQnapus2000	HI_H	3.72	13.64
A05	4109774	** *UQnapus2563* **	HI_H	4.58	15.90
A05	10231305	UQnapus5012	BM_H	3.24	10.34
A05	17371711	UQnapus0273	Y5P_H	3.18	10.80
A06	3956415	UQnapus5652	BM_H	3.33	10.46
A07	6234486	UQnapus2420	HI_H	4.00	14.11
A07	11614849	** *UQnapus2473* **	Y5P_H	4.69	15.67
A07	17122140	UQnapus0069	Y5P_H	5.16	9.23
A08	2617392	UQnapus2637	Y5P_H	3.80	13.02
A09	8745612	UQnapus2826	YWP_H	3.03	8.04
A09	36005032	UQnapus2957	Y5P_H	3.14	10.93
A10	6601391	** *UQnapus4615* **	BM_H	5.04	15.61
A10	8771845	UQnapus3070	BM_H	3.13	8.86
Under control condition					
A01	6681985	UQnapus5365	BM_C	3.52	14.03
A01	14706490	UQnapus1205	Y5P_C	3.94	12.20
A01	16284432	UQnapus3105	BM_C	3.08	11.89
A01	20392779	UQnapus1241	HI_C	3.81	12.05
A02	1250723	UQnapus5063	BM_C	3.85	7.05
A02	8703496	UQnapus1394	YWP_C	3.12	8.64
A02	8884662	UQnapus5442	YWP_C	3.08	10.53
A02	8949907	UQnapus1398	HI_C	3.57	11.64
A03	5898488	** *UQnapus1557* **	BM_C	3.67	13.07
A03	16861693	UQnapus1638	HI_C	7.46	22.77
A04	11451264	** *UQnapus1923* **	HI_C	3.68	12.20
A05	4109774	** *UQnapus2563* **	HI_C	6.31	20.35
A05	17680875	UQnapus2144	YWP_C	3.38	9.73
A06	1530076	UQnapus2220	BM_C	3.23	13.05
A06	2902967	UQnapus5649	YWP_C	3.79	11.01
A06	3186312	UQnapus5341	HI_C	3.84	13.01
A06	15891262	UQnapus5014	HI_C	3.53	12.31
A07	9119865	UQnapus2455	BM_C	3.43	13.20
A07	11614849	** *UQnapus2473* **	YWP_C	3.71	10.81
A07	17157679	UQnapus0091	YWP_C	3.22	10.48
A08	8043536	UQnapus2097	Y5P_C	3.83	19.78
A08	20418844	UQnapus2762	YWP_C	3.01	8.85
A09	798560	** *UQnapus2777* **	HI_C	7.46	22.82
A09	798560	** *UQnapus2777* **	YWP_C	3.19	9.04
A09	1372115	UQnapus2783	YWP_C	3.20	10.48
A09	6624646	UQnapus2806	YWP_C	3.29	10.48
A09	7086176	UQnapus2807	HI_C	3.13	10.25
A09	7452134	UQnapus2809	YWP_C	3.34	9.13
A09	27135256	UQnapus2916	HI_C	4.30	15.47
A09	35718978	UQnapus3633	HI_C	4.16	23.43
A10	4433065	UQnapus1779	YWP_C	3.84	11.94
A10	6507394	UQnapus1763	HI_C	3.03	10.47
A10	6601391	** *UQnapus4615* **	BM_C	4.65	16.14

**Table 5 genes-13-00296-t005:** Nine candidate genes were identified on a genome region of 225 kb flanking on both sides of UQnapus2473 on Chromosome A07.

#	*B. rapa*	*B. napus*	At Ortholog	Gene	Annotation/Function (GO/At Ortholog Function)
1	Bra015593	BnaA07g01650D	AT3G48800.1	Sterile α motif (SAM) domain-containing protein	Sterile α motif (SAM) domain-containing protein
2	Bra038828	BnaA07g01660D	AT2G18890.1	Protein kinase superfamily protein	Protein kinase superfamily protein; FUNCTIONS IN: protein serine/threonine kinase activity, protein kinase activity, ATP binding; INVOLVED IN: protein amino acid phosphorylation
3		BnaA07g01670D	AT2G15740.1 AT5G42640.1	C2H2 and C2H2-like zinc fingers superfamily protein	FUNCTIONS IN: sequence-specific DNA binding transcription factor activity, zinc ion binding, nucleic acid binding; INVOLVED IN: regulation of transcription; LOCATED IN: intracellular
4	Bra039606	BnaA07g01680D	AT2G18490.1 AT2G15740.1 AT5G42640.1		
5	Bra039605	BnaA07g01690D	AT5G09920.1	RNA polymerase II, Rpb4, core protein	Non-catalytic subunit specific to DNA-dependent RNA polymerase II; the ortholog of budding yeast RPB4
6	Bra039604	BnaA07g01700D	AT2G18500.1	Ovate family protein 7 (OFP7)	INVOLVED IN: biological process unknown; LOCATED IN: plasma membrane; EXPRESSED IN: shoot apex, embryo, hypocotyl, flower, seed
7	Bra039603	BnaA07g01710D	AT2G18510.1	Embryo defective 2444 (emb2444), RNA-binding (RRM/RBD/RNP motifs) family protein	FUNCTIONS IN: RNA binding, nucleotide binding, nucleic acid binding; INVOLVED IN: embryo development ending in seed dormancy; LOCATED IN: nucleolus
8	Bra028100	BnaA07g01720D	AT4G36710.1	GRAS family transcription factor	CONTAINS InterPro DOMAIN/s: Transcription factor GRAS (InterPro:IPR005202)
9	Bra028098	BnaA07g01730D	AT2G18570.1 AT2G18560.1	UDP-Glycosyltransferase superfamily protein	FUNCTIONS IN: transferase activity, transferring glycosyl groups; INVOLVED IN: metabolic process; LOCATED IN: cellular component unknown

## Data Availability

Data is contained within the article and supplementary material.

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
