# Peer review of "Quantitative Trait Loci for Heat Stress Tolerance in Brassica rapa L. Are Distributed across the Genome and Occur in Diverse Genetic Groups, Flowering Phenologies and Morphotypes"

_genes, 2022, doi:10.3390/genes13020296_

Round 1
Reviewer 1 Report
This manuscript describes an elegant study that has identified several heat-tolerant Brassica rapa lines, which maintained grain yield, biomass and harvest index after 7 days of heat stress at the first flower. The authors provide a large body of data in support of the conclusion that QTLs for heat stress tolerance in B. rapa are distributed across the genome and occur in diverse genetic groups, flowering phenologies and morphotypes.
The sensitivity of different Brassica crops to heat stress during flowering is well documented. Similarly, genetic variation in sensitivity to heat stress has been reported in B. rapa accessions. The study reported in this manuscript builds on this firm foundation. Using a genetically diverse global collection of B. rapa germplasm, the authors have evaluated nine traits for heat stress tolerance using simulated transient daily heat stress after the first open flower in well-watered plants. Plants were grown in pots under appropriate light, temperature and humidity conditions. Plants were removed to controlled environment rooms (CER) and different treatments were applied. CER1 (control: 25 °C day/ 15 °C night: CER2 (heat stress: 35 °C). It is unfortunate that the irradiance in the CER was substantially less than that in the greenhouse. There was decrease from 635 mmol m–2 s –1 prior to 425 mmol m–2 s –1 light. One presumes that the plants were allowed to acclimate to the new light environment prior to the heat stress treatments, but this is not mentioned in the Materials and methods section. Can the authors explain more clearly how the experiment was performed because a substantial change in irradiance will change plant responses to environmental stresses?
Heat-sensitive lines were shown to have a significant reduction in yield-related traits following the heat stress treatment compared to the controls. Heat-tolerant lines were found among all morphotypes (leafy, rooty and oilseed) and flowering phenologies (spring, winter and semi-winter types). The authors identified nine candidate genes associated with significant SNPs which may contribute to heat tolerance.
This manuscript describes an elegant study that has identified several heat-tolerant Brassica rapa lines, which maintained grain yield, biomass and harvest index after 7 days of heat stress at the first flower. The authors provide a large body of data in support of the conclusion that QTLs for heat stress tolerance in B. rapa are distributed across the genome and occur in diverse genetic groups, flowering phenologies and morphotypes.
The sensitivity of different Brassica crops to heat stress during flowering is well documented. Similarly, genetic variation in sensitivity to heat stress has been reported in B. rapa accessions. The study reported in this manuscript builds on this firm foundation. Using a genetically diverse global collection of B. rapa germplasm, the authors have evaluated nine traits for heat stress tolerance using simulated transient daily heat stress after the first open flower in well-watered plants. Plants were grown in pots under appropriate light, temperature and humidity conditions. Plants were removed to controlled environment rooms (CER) and different treatments were applied. CER1 (control: 25 °C day/ 15 °C night: CER2 (heat stress: 35 °C). It is unfortunate that the irradiance in the CER was substantially less than that in the greenhouse. There was decrease from 635 mmol m–2 s –1 prior to 425 mmol m–2 s –1 light. One presumes that the plants were allowed to acclimate to the new light environment prior to the heat stress treatments, but this is not mentioned in the Materials and methods section. Can the authors explain more clearly how the experiment was performed because a substantial change in irradiance will change plant responses to environmental stresses?
Heat-sensitive lines were shown to have a significant reduction in yield-related traits following the heat stress treatment compared to the controls. Heat-tolerant lines were found among all morphotypes (leafy, rooty and oilseed) and flowering phenologies (spring, winter and semi-winter types). The authors identified nine candidate genes associated with significant SNPs which may contribute to heat tolerance. Overall, the data are clearly presented and appropriately discussed. The findings are important and useful to a wide community of researchers interested in enhancing heat tolerance traits in crop species.
This manuscript describes an elegant study that has identified several heat-tolerant Brassica rapa lines, which maintained grain yield, biomass and harvest index after 7 days of heat stress at the first flower. The authors provide a large body of data in support of the conclusion that QTLs for heat stress tolerance in B. rapa are distributed across the genome and occur in diverse genetic groups, flowering phenologies and morphotypes.
The sensitivity of different Brassica crops to heat stress during flowering is well documented. Similarly, genetic variation in sensitivity to heat stress has been reported in B. rapa accessions. The study reported in this manuscript builds on this firm foundation. Using a genetically diverse global collection of B. rapa germplasm, the authors have evaluated nine traits for heat stress tolerance using simulated transient daily heat stress after the first open flower in well-watered plants. Plants were grown in pots under appropriate light, temperature and humidity conditions. Plants were removed to controlled environment rooms (CER) and different treatments were applied. CER1 (control: 25 °C day/ 15 °C night: CER2 (heat stress: 35 °C). It is unfortunate that the irradiance in the CER was substantially less than that in the greenhouse. There was decrease from 635 mmol m–2 s –1 prior to 425 mmol m–2 s –1 light. One presumes that the plants were allowed to acclimate to the new light environment prior to the heat stress treatments, but this is not mentioned in the Materials and methods section. Can the authors explain more clearly how the experiment was performed because a substantial change in irradiance will change plant responses to environmental stresses?
Heat-sensitive lines were shown to have a significant reduction in yield-related traits following the heat stress treatment compared to the controls. Heat-tolerant lines were found among all morphotypes (leafy, rooty and oilseed) and flowering phenologies (spring, winter and semi-winter types). The authors identified nine candidate genes associated with significant SNPs which may contribute to heat tolerance. Overall, the data are clearly presented and appropriately discussed. The findings are important and useful to a wide community of researchers interested in enhancing heat tolerance traits in crop species.
This manuscript describes an elegant study that has identified several heat-tolerant Brassica rapa lines, which maintained grain yield, biomass and harvest index after 7 days of heat stress at the first flower. The authors provide a large body of data in support of the conclusion that QTLs for heat stress tolerance in B. rapa are distributed across the genome and occur in diverse genetic groups, flowering phenologies and morphotypes.
The sensitivity of different Brassica crops to heat stress during flowering is well documented. Similarly, genetic variation in sensitivity to heat stress has been reported in B. rapa accessions. The study reported in this manuscript builds on this firm foundation. Using a genetically diverse global collection of B. rapa germplasm, the authors have evaluated nine traits for heat stress tolerance using simulated transient daily heat stress after the first open flower in well-watered plants. Plants were grown in pots under appropriate light, temperature and humidity conditions. Plants were removed to controlled environment rooms (CER) and different treatments were applied. CER1 (control: 25 °C day/ 15 °C night: CER2 (heat stress: 35 °C). It is unfortunate that the irradiance in the CER was substantially less than that in the greenhouse. There was decrease from 635 mmol m–2 s –1 prior to 425 mmol m–2 s –1 light. One presumes that the plants were allowed to acclimate to the new light environment prior to the heat stress treatments, but this is not mentioned in the Materials and methods section. Can the authors explain more clearly how the experiment was performed because a substantial change in irradiance will change plant responses to environmental stresses?
Heat-sensitive lines were shown to have a significant reduction in yield-related traits following the heat stress treatment compared to the controls. Heat-tolerant lines were found among all morphotypes (leafy, rooty and oilseed) and flowering phenologies (spring, winter and semi-winter types). The authors identified nine candidate genes associated with significant SNPs which may contribute to heat tolerance. Overall, the data are clearly presented and appropriately discussed. The findings are important and useful to a wide community of researchers interested in enhancing heat tolerance traits in crop species.
This manuscript describes an elegant study that has identified several heat-tolerant Brassica rapa lines, which maintained grain yield, biomass and harvest index after 7 days of heat stress at the first flower. The authors provide a large body of data in support of the conclusion that QTLs for heat stress tolerance in B. rapa are distributed across the genome and occur in diverse genetic groups, flowering phenologies and morphotypes.
The sensitivity of different Brassica crops to heat stress during flowering is well documented. Similarly, genetic variation in sensitivity to heat stress has been reported in B. rapa accessions. The study reported in this manuscript builds on this firm foundation. Using a genetically diverse global collection of B. rapa germplasm, the authors have evaluated nine traits for heat stress tolerance using simulated transient daily heat stress after the first open flower in well-watered plants. Plants were grown in pots under appropriate light, temperature and humidity conditions. Plants were removed to controlled environment rooms (CER) and different treatments were applied. CER1 (control: 25 °C day/ 15 °C night: CER2 (heat stress: 35 °C). It is unfortunate that the irradiance in the CER was substantially less than that in the greenhouse. There was decrease from 635 mmol m–2 s –1 prior to 425 mmol m–2 s –1 light. One presumes that the plants were allowed to acclimate to the new light environment prior to the heat stress treatments, but this is not mentioned in the Materials and methods section. Can the authors explain more clearly how the experiment was performed because a substantial change in irradiance will change plant responses to environmental stresses?
Heat-sensitive lines were shown to have a significant reduction in yield-related traits following the heat stress treatment compared to the controls. Heat-tolerant lines were found among all morphotypes (leafy, rooty and oilseed) and flowering phenologies (spring, winter and semi-winter types). The authors identified nine candidate genes associated with significant SNPs which may contribute to heat tolerance. Overall, the data are clearly presented and appropriately discussed. The findings are important and useful to a wide community of researchers interested in enhancing heat tolerance traits in crop species.
This manuscript describes an elegant study that has identified several heat-tolerant Brassica rapa lines, which maintained grain yield, biomass and harvest index after 7 days of heat stress at the first flower. The authors provide a large body of data in support of the conclusion that QTLs for heat stress tolerance in B. rapa are distributed across the genome and occur in diverse genetic groups, flowering phenologies and morphotypes.
The sensitivity of different Brassica crops to heat stress during flowering is well documented. Similarly, genetic variation in sensitivity to heat stress has been reported in B. rapa accessions. The study reported in this manuscript builds on this firm foundation. Using a genetically diverse global collection of B. rapa germplasm, the authors have evaluated nine traits for heat stress tolerance using simulated transient daily heat stress after the first open flower in well-watered plants. Plants were grown in pots under appropriate light, temperature and humidity conditions. Plants were removed to controlled environment rooms (CER) and different treatments were applied. CER1 (control: 25 °C day/ 15 °C night: CER2 (heat stress: 35 °C). It is unfortunate that the irradiance in the CER was substantially less than that in the greenhouse. There was decrease from 635 mmol m–2 s –1 prior to 425 mmol m–2 s –1 light. One presumes that the plants were allowed to acclimate to the new light environment prior to the heat stress treatments, but this is not mentioned in the Materials and methods section. Can the authors explain more clearly how the experiment was performed because a substantial change in irradiance will change plant responses to environmental stresses?
Heat-sensitive lines were shown to have a significant reduction in yield-related traits following the heat stress treatment compared to the controls. Heat-tolerant lines were found among all morphotypes (leafy, rooty and oilseed) and flowering phenologies (spring, winter and semi-winter types). The authors identified nine candidate genes associated with significant SNPs which may contribute to heat tolerance. Overall, the data are clearly presented and appropriately discussed. The findings are important and useful to a wide community of researchers interested in enhancing heat tolerance traits in crop species.
This manuscript describes an elegant study that has identified several heat-tolerant Brassica rapa lines, which maintained grain yield, biomass and harvest index after 7 days of heat stress at the first flower. The authors provide a large body of data in support of the conclusion that QTLs for heat stress tolerance in B. rapa are distributed across the genome and occur in diverse genetic groups, flowering phenologies and morphotypes.
The sensitivity of different Brassica crops to heat stress during flowering is well documented. Similarly, genetic variation in sensitivity to heat stress has been reported in B. rapa accessions. The study reported in this manuscript builds on this firm foundation. Using a genetically diverse global collection of B. rapa germplasm, the authors have evaluated nine traits for heat stress tolerance using simulated transient daily heat stress after the first open flower in well-watered plants. Plants were grown in pots under appropriate light, temperature and humidity conditions. Plants were removed to controlled environment rooms (CER) and different treatments were applied. CER1 (control: 25 °C day/ 15 °C night: CER2 (heat stress: 35 °C). It is unfortunate that the irradiance in the CER was substantially less than that in the greenhouse. There was decrease from 635 mmol m–2 s –1 prior to 425 mmol m–2 s –1 light. One presumes that the plants were allowed to acclimate to the new light environment prior to the heat stress treatments, but this is not mentioned in the Materials and methods section. Can the authors explain more clearly how the experiment was performed because a substantial change in irradiance will change plant responses to environmental stresses?
Heat-sensitive lines were shown to have a significant reduction in yield-related traits following the heat stress treatment compared to the controls. Heat-tolerant lines were found among all morphotypes (leafy, rooty and oilseed) and flowering phenologies (spring, winter and semi-winter types). The authors identified nine candidate genes associated with significant SNPs which may contribute to heat tolerance. Overall, the data are clearly presented and appropriately discussed. The findings are important and useful to a wide community of researchers interested in enhancing heat tolerance traits in crop species.
This manuscript describes an elegant study that has identified several heat-tolerant Brassica rapa lines, which maintained grain yield, biomass and harvest index after 7 days of heat stress at the first flower. The authors provide a large body of data in support of the conclusion that QTLs for heat stress tolerance in B. rapa are distributed across the genome and occur in diverse genetic groups, flowering phenologies and morphotypes.
The sensitivity of different Brassica crops to heat stress during flowering is well documented. Similarly, genetic variation in sensitivity to heat stress has been reported in B. rapa accessions. The study reported in this manuscript builds on this firm foundation. Using a genetically diverse global collection of B. rapa germplasm, the authors have evaluated nine traits for heat stress tolerance using simulated transient daily heat stress after the first open flower in well-watered plants. Plants were grown in pots under appropriate light, temperature and humidity conditions. Plants were removed to controlled environment rooms (CER) and different treatments were applied. CER1 (control: 25 °C day/ 15 °C night: CER2 (heat stress: 35 °C). It is unfortunate that the irradiance in the CER was substantially less than that in the greenhouse. There was decrease from 635 mmol m–2 s –1 prior to 425 mmol m–2 s –1 light. One presumes that the plants were allowed to acclimate to the new light environment prior to the heat stress treatments, but this is not mentioned in the Materials and methods section. Can the authors explain more clearly how the experiment was performed because a substantial change in irradiance will change plant responses to environmental stresses?
Heat-sensitive lines were shown to have a significant reduction in yield-related traits following the heat stress treatment compared to the controls. Heat-tolerant lines were found among all morphotypes (leafy, rooty and oilseed) and flowering phenologies (spring, winter and semi-winter types). The authors identified nine candidate genes associated with significant SNPs which may contribute to heat tolerance. Overall, the data are clearly presented and appropriately discussed. The findings are important and useful to a wide community of researchers interested in enhancing heat tolerance traits in crop species.
This manuscript describes an elegant study that has identified several heat-tolerant Brassica rapa lines, which maintained grain yield, biomass and harvest index after 7 days of heat stress at the first flower. The authors provide a large body of data in support of the conclusion that QTLs for heat stress tolerance in B. rapa are distributed across the genome and occur in diverse genetic groups, flowering phenologies and morphotypes.
The sensitivity of different Brassica crops to heat stress during flowering is well documented. Similarly, genetic variation in sensitivity to heat stress has been reported in B. rapa accessions. The study reported in this manuscript builds on this firm foundation. Using a genetically diverse global collection of B. rapa germplasm, the authors have evaluated nine traits for heat stress tolerance using simulated transient daily heat stress after the first open flower in well-watered plants. Plants were grown in pots under appropriate light, temperature and humidity conditions. Plants were removed to controlled environment rooms (CER) and different treatments were applied. CER1 (control: 25 °C day/ 15 °C night: CER2 (heat stress: 35 °C). It is unfortunate that the irradiance in the CER was substantially less than that in the greenhouse. There was decrease from 635 mmol m–2 s –1 prior to 425 mmol m–2 s –1 light. One presumes that the plants were allowed to acclimate to the new light environment prior to the heat stress treatments, but this is not mentioned in the Materials and methods section. Can the authors explain more clearly how the experiment was performed because a substantial change in irradiance will change plant responses to environmental stresses?
Heat-sensitive lines were shown to have a significant reduction in yield-related traits following the heat stress treatment compared to the controls. Heat-tolerant lines were found among all morphotypes (leafy, rooty and oilseed) and flowering phenologies (spring, winter and semi-winter types). The authors identified nine candidate genes associated with significant SNPs which may contribute to heat tolerance. Overall, the data are clearly presented and appropriately discussed. The findings are important and useful to a wide community of researchers interested in enhancing heat tolerance traits in crop species.
This manuscript describes an elegant study that has identified several heat-tolerant Brassica rapa lines, which maintained grain yield, biomass and harvest index after 7 days of heat stress at the first flower. The authors provide a large body of data in support of the conclusion that QTLs for heat stress tolerance in B. rapa are distributed across the genome and occur in diverse genetic groups, flowering phenologies and morphotypes.
The sensitivity of different Brassica crops to heat stress during flowering is well documented. Similarly, genetic variation in sensitivity to heat stress has been reported in B. rapa accessions. The study reported in this manuscript builds on this firm foundation. Using a genetically diverse global collection of B. rapa germplasm, the authors have evaluated nine traits for heat stress tolerance using simulated transient daily heat stress after the first open flower in well-watered plants. Plants were grown in pots under appropriate light, temperature and humidity conditions. Plants were removed to controlled environment rooms (CER) and different treatments were applied. CER1 (control: 25 °C day/ 15 °C night: CER2 (heat stress: 35 °C). It is unfortunate that the irradiance in the CER was substantially less than that in the greenhouse. There was decrease from 635 mmol m–2 s –1 prior to 425 mmol m–2 s –1 light. One presumes that the plants were allowed to acclimate to the new light environment prior to the heat stress treatments, but this is not mentioned in the Materials and methods section. Can the authors explain more clearly how the experiment was performed because a substantial change in irradiance will change plant responses to environmental stresses?
Heat-sensitive lines were shown to have a significant reduction in yield-related traits following the heat stress treatment compared to the controls. Heat-tolerant lines were found among all morphotypes (leafy, rooty and oilseed) and flowering phenologies (spring, winter and semi-winter types). The authors identified nine candidate genes associated with significant SNPs which may contribute to heat tolerance. Overall, the data are clearly presented and appropriately discussed. The findings are important and useful to a wide community of researchers interested in enhancing heat tolerance traits in crop species.
This manuscript describes an elegant study that has identified several heat-tolerant Brassica rapa lines, which maintained grain yield, biomass and harvest index after 7 days of heat stress at the first flower. The authors provide a large body of data in support of the conclusion that QTLs for heat stress tolerance in B. rapa are distributed across the genome and occur in diverse genetic groups, flowering phenologies and morphotypes.
The sensitivity of different Brassica crops to heat stress during flowering is well documented. Similarly, genetic variation in sensitivity to heat stress has been reported in B. rapa accessions. The study reported in this manuscript builds on this firm foundation. Using a genetically diverse global collection of B. rapa germplasm, the authors have evaluated nine traits for heat stress tolerance using simulated transient daily heat stress after the first open flower in well-watered plants. Plants were grown in pots under appropriate light, temperature and humidity conditions. Plants were removed to controlled environment rooms (CER) and different treatments were applied. CER1 (control: 25 °C day/ 15 °C night: CER2 (heat stress: 35 °C). It is unfortunate that the irradiance in the CER was substantially less than that in the greenhouse. There was decrease from 635 mmol m–2 s –1 prior to 425 mmol m–2 s –1 light. One presumes that the plants were allowed to acclimate to the new light environment prior to the heat stress treatments, but this is not mentioned in the Materials and methods section. Can the authors explain more clearly how the experiment was performed because a substantial change in irradiance will change plant responses to environmental stresses?
Heat-sensitive lines were shown to have a significant reduction in yield-related traits following the heat stress treatment compared to the controls. Heat-tolerant lines were found among all morphotypes and flowering phenologies. The authors identified nine candidate genes associated with significant SNPs which may contribute to heat tolerance. Overall, the data are clearly presented and appropriately discussed. The findings are important and useful to a wide community of researchers interested in enhancing heat tolerance traits in crop species.
Overall, the data are clearly presented and appropriately discussed. The findings are important and useful to a wide community of researchers interested in enhancing heat tolerance traits in crop species.
Author Response
Point 1: It is unfortunate that the irradiance in the CER was substantially less than that in the greenhouse. There was decrease from 635 mmol m–2 s –1 prior to 425 mmol m–2 s –1 light. One presumes that the plants were allowed to acclimate to the new light environment prior to the heat stress treatments, but this is not mentioned in the Materials and methods section. Can the authors explain more clearly how the experiment was performed because a substantial change in irradiance will change plant responses to environmental stresses?
Response 1: We thank the reviewer for pointing out this. Light readings in the glasshouse were restricted to mid-day on sunny days, and of course light intensity varied in the glasshouse during the day depending on cloud, time of day, time of year, and so on. We set out very carefully in this experiment to make sure that the two Sets of plants received the same conditions for all inputs, such as light intensity, day-length, nutrients, adequate water, and so on, except for 7 days of temperature treatment at first flower – and the high temperature and control temperature treatments in the CERs were under the same conditions of light, water, nutrients, day-length etc except for temperature. We controlled the experiment appropriately to meet the requirements of a controlled scientific experiment on the impact of heat stress on the first 7 days of the reproductive phase. We appreciate that light intensity could be a valuable variable to follow up in heat experiments, but that was not the goal of this experiment. All treatments had the same light conditions.
We have added the following comment to Methods (Lines 94-95): “Light readings in the glasshouse were taken at mid-day on a sunny day in mid-spring in an unshaded area of the glasshouse.” And, after describing the CER conditions (Lines 113-119): “Both high temperature and control temperature CER’s had the same light intensity, same day-length, same humidity, same nutrition and adequate water to avoid any confounding of heat stress treatment with other effects that might have differed between the treatments.
Both Sets of plants, destined for high and control temperature treatments, received the same growth conditions before, during and after the temperature treatments to avoid any confounding of heat stress with other growth inputs such as water, nutrients and light.”

Reviewer 2 Report
The identification of QTLs for heat tolerance is of utmost value in the era of global warming.
The manuscript is of great scientific value. QTL identification is important for developing heat-resistant cultivars in heat-sensitive crops.
Just need to address minor points as found in the attached file.

Author Response
Point 1: L81-82 Why not all the 217 lines were subjected to phenotyping (physiological and yield traits) and genotyping (SNP based)?
Response 1: We thank the reviewer for this good point. Actually all the 217 lines were subjected to SNP genotyping, and the average call rate of all lines was 87.04%. There were 5 lines with their call rate below 70.0% and they were excluded for further SNP genetic diversity analysis and GWAS analysis because of the low call rate. Now we have added in Lines 171-172 of the revised version (Method section): “Samples with their call rate below 70% were excluded for further data analysis” and also in Lines 278-280 (Results section): “All 217 lines were subjected to SNP genotyping, and the average call rate of all lines was 87.04%, and 5 lines with their call rate below 70.0% were excluded for further data analysis.”
Of the 217 lines, 75 lines with limited number of seeds and with self-incompatibility were excluded for phenotyping experiment, so finally 142 lines were phenotyped for several physiological and yield-related traits. We have also added it in Line 82-83 of the revised version.
Point 2: L83 Control is normal growing condition?
Response 2: Yes, “Control” treatment was based on normal growing conditions, with no heat stress. Now we have clarified it in the revised version.
Reviewer 3 Report
Dear editor,
I would like to appreciate for giving me the chance to review the draft titled “Quantitative trait loci for heat stress tolerance in Brassica rapa L. are distributed across the genome and occur in diverse genetic groups, flowering phenologies and morphotypes” by Sheng et al. during this research activity, authors tried to induce heat stress on different accession of B. rapa and evaluate the physiological and molecular genetic diversity of the plants. The results of the current experiment confirmed two major genetic populations; one from East and South Asia and one from Europe. Additionally, genome-wide asociation analysis of heat stress-related yield traits revealed 57 SNPs distributed across all 10 B. rapa chromosomes, some of which were associated with potential candidate genes for heat stress tolerance.
The process of preparation is confusing for me. The number of plants in each set hasn’t mentioned by authors and the statistical design is not clear. As I understood, authors used plants for this experiment. I would like to know the age of plants, phenotypical stage of plants, even if the plants were uniform? Number of leaves and branches were same? Even I would like to know if the acclimatization were done before the experiment or not? On the other hand, why authors used plants rather than seeds? I appreciate it if the author makes clear the above matters for the readers.
Why did authors measure the physiological traits on the day 7th? What is the milestone?
Best wishes
Dear editor,
I would like to appreciate for giving me the chance to review the draft titled “Quantitative trait loci for heat stress tolerance in Brassica rapa L. are distributed across the genome and occur in diverse genetic groups, flowering phenologies and morphotypes” by Sheng et al. during this research activity, authors tried to induce heat stress on different accession of B. rapa and evaluate the physiological and molecular genetic diversity of the plants. The results of the current experiment confirmed two major genetic populations; one from East and South Asia and one from Europe. Additionally, genome-wide association analysis of heat stress-related yield traits revealed 57 SNPs distributed across all 10 B. rapa chromosomes, some of which were associated with potential candidate genes for heat stress tolerance. I would like to suggest the mentioned draft for publication with the minor correction as follow:
The process of preparation is confusing for me. The number of plants in each set hasn’t been mentioned by the authors and the statistical design is not clear. As I understood, the authors used plants for this experiment. I would like to know the age of plants, phenotypical stage of plants, even if the plants were uniform? Number of leaves and branches were same? Even I would like to know if the acclimatization were done before the experiment or not? On the other hand, why do authors use plants rather than seeds? I appreciate it if the author makes clear the above matters for the readers.
Why did the authors measure the physiological traits on day 7th? What is the milestone?
Best wishes
Author Response
Response 1:
We thank the reviewer for pointing out that we need to make our methods more clear. We have revised the section on the number of plants in each Set at Lines 87-90 of the revised version:
“Two Sets of plants were sown in pots, with a single plant in each pot. Set 1 plants were grown for control (no heat stress) treatment, and Set 2 plants were grown for heat stress treatment at first flower on the main stem. There were three replications (pots) in each Set.”
We should clarify that our experiments were based on heat stress at the most sensitive stage to heat stress in Brassica species (at first flower) so we have revised Lines 65-67 of Introduction:
“Based on our previous experience that heat stress has its greatest impact during the reproductive stage in B. rapa and B. napus [4, 13, 14, 17], we evaluated a genetically diverse global collection of B. rapa germplasm for heat stress tolerance following simulated transient daily heat stress after the first open flower on the main stem of well-watered plants.”
In our research, the focus of the heat stress treatment was on first few days after flowering began on the main stem. This is clearly described in Methods (Lines 100-102):
“When the first open flower was seen on the main stem, plants were moved to a controlled environment room (CER) for seven days of temperature treatment ...”
That is, all plants were evaluated at the same phenological stage (first flowering on the main stem), and all plants were grown under the same moderate glasshouse growth conditions before first flower. No special “acclimatisation” occurred, and all genotypes flowered without any special treatments. The number of days from sowing to first flower (DTF) varied widely among this diverse group of accessions, as would all the other attributes of plant growth associated with date of first flower. We have added one new column “DTF” in the revised Supplementary Table S1.
This research was focussed on the impact of heat stress during first flowering and therefore we did not measure number of leaves, branches or other factors in growth because this was not relevant to the measures of heat stress tolerance. However, we did measure final biomass, final grain yield and harvest index as a guide to plant growth after heat stress during first flower.
The reviewer asks why we used plants rather than seeds. Heat stress tolerance in seeds may be an issue in some environments, but it is irrelevant to our study. As reported in the large accumulated literature cited in the Introduction and Discussion in this paper, heat stress has its greatest impact on grain-producing Brassica crops during the reproductive stage. In fact, in our results in this study, we show that heat stress treatment did not reduce (but increased) plant biomass, so heat stress treatment for 7 d during flowering actually had a beneficial effect on biomass. However, seed development on the main stem, whole plant grain yield and harvest index was severely inhibited in heat-sensitive lines.
Point 2: Why did authors measure the physiological traits on the day 7th? What is the milestone?
Response 2: In previous studies (13, 14), we reported more frequent measures of physiological traits during heat stress treatment, and on the basis of these previous results we decided to restrict the number of measurements of physiological traits in this study to day 7 of heat stress treatment. On the basis of our experience, we decided that there would be little negative impact of this decision on the results and conclusions of this study.
